# Urine HPV in the Context of Genital and Cervical Cancer Screening—An Update of Current Literature

**DOI:** 10.3390/cancers13071640

**Published:** 2021-04-01

**Authors:** Alexandros Daponte, George Michail, Athina-Ioanna Daponte, Nikoletta Daponte, George Valasoulis

**Affiliations:** 1Department of Obstetrics & Gynaecology, University Hospital of Larisa, 41334 Larisa, Greece; atdaponte@uth.gr (A.-I.D.); daponte@uth.gr (N.D.); obgynmaster@uth.gr (G.V.); 2Department of Obstetrics & Gynaecology, University Hospital of Patras, 26504 Patras, Greece; gmichail@upatras.gr; 3Hellenic National Public Health Organization—ECDC, Marousi, 15123 Athens, Greece

**Keywords:** HPV, HPV urine, cervical cancer screening, HPV DNA, genotyping, mRNA E6 & E7, HPV methylation markers

## Abstract

**Simple Summary:**

Despite the substantial scientific evolution in cervical cancer prevention and related infrastructures, a plethora of women still miss the opportunity to detect their precancerous lesions at a curable stage by not participating in existing screening programs. Implementing sensitive screening modalities combined with easy sampling methods with minimal pain or discomfort such as self-sampling of vaginal and urine samples is increasingly applied. Self-sampling HPV modalities aimed to address this inequity, besides facilitating HPV genotyping as well as the measurement of related biomarkers in HPV-caused lesions and genital cancer. The low costs inflicted, the non-invasive nature, and the favorable acceptability profile of urine HPV detection give the potential to become a most promising tool that could expand the possibilities in changing genital and cervical cancer prevention strategies as well as in the surveillance and management of genital precancer.

**Abstract:**

Within the previous decades, following the widespread implementation of HPV-related biomarkers and computerization in liquid-based cytology, screening for lower genital tract malignancies has been optimized in several parts of the world. Many organized anogenital cancer prevention systems have reached a point at which efficacy is more a matter of population coverage and less of available infrastructures. Meanwhile, self-sampling modalities in which biologic material (vaginal secretions, urine, etc.) is obtained by the individual and not the clinician and subsequently undergoes examination for HPV biomarkers enjoy appreciating acceptance. Bygone the initial skepticism that vaginal or urine HPV represents “passenger” transient infections, extensive scientific work has been conducted to optimize high-risk HPV (hrHPV) detection from this “novel” biologic material. Nowadays, several state-of-the-art meta-analyses have illustrated that self-sampling techniques involving urine self-sampling represent a feasible alternative strategy with potentially enhanced population coverage possessing excellent performance and sensitivity. Recently published scientific work focusing on urine HPV was reviewed, and after a critical appraisal, the following points should be considered in the clinical application of hrHPV urine measurements; (i) use of first-void urine (FVU) and purpose-designed collection devices; (ii) using a preservation medium to avoid human/HPV DNA degradation during extraction and storage; (iii) using polymerase chain reaction (PCR) based assays, ideally with genotyping capabilities; (iv) processing of a sufficient volume of whole urine; and (v) the use of an analytically sensitive HPV test/recovery of cell-free HPV DNA in addition to cell-associated DNA.

## 1. Introduction

Despite the substantial scientific evolution in cervical cancer prevention and related infrastructures, millions of “at-risk” women miss the opportunity to detect their precancerous lesions at a curable stage by not participating in screening programs. Sometimes, this happens because a large proportion of the population is lost to cervical cancer screening due to poor resources, cultural barriers, or avoidance of a pelvic exam [1]. Additionally, sometimes these women live in low-resource settings or belong to social or religious minorities; however, a considerable percentage of cervical screening non-attenders simply feel uncomfortable with the current screening protocols and methodology, including visiting health facilities, more so during the COVID-19 pandemic, and/or trying to avoid the gynecological examination and the lithotomy position [2,3].

During the previous decades, the progressive implementation of HPV-related cervical screenings has transformed the whole paradigm in cervical cancer prevention. HPV-based molecular screening modalities proved to be highly sensitive, thus offering better protection from cervical cancer than cytology [4,5,6,7,8,9]. However, the necessity for optimizing the cost-effectiveness of screening interventions is paramount and seems now even more justified, with health systems’ resources confronted globally by unexpected groundbreaking challenges, like the COVID-19 pandemic [10]. In the quest of increasing genital and cervical screening effectiveness, suboptimal participation rates of the targeted population in screening programs represent a fundamental drawback [11]. An important immediate gain, not only in cervical cancer but also in cancer prevention as a whole, could be attained by increasing screening the attendance among women who are currently either unscreened or screened infrequently using self-sampling HPV detection methods [4,12,13,14,15,16,17], given the causative role of HPV in a plethora of cancers. Implementing sensitive screening modalities combined with easy sampling methods with minimal pain or discomfort bypassing the doctors’ visit, such as self-sampling of vaginal and urine samples, is feasible and possibly cost-effective [18].

Older and recent studies have illustrated that self-sampling is highly acceptable among women [6,12,13,14], even during pregnancy [19] or infertility work up [20]. This popularity could be partially attributed to overcoming hesitance and concerns about disclosing sexual activity history. Acceptance rates could possibly be further improved with proper communication of the procedure and documentation of its non-inferiority compared with conventional screening [6].

Persistent high-risk HPV (hrHPV) infections are the leading cause of more than 90% of cervical and anal cancers, approximately 70% of vaginal and vulvar cancers, and more than 50% of penile cancers [21]. Based on data from 2013 to 2017, about 45,300 HPV-related neoplasms occur in the United States each year: about 25,400 among women and about 19,900 among men. Cervical cancer remains the most common HPV-related disease among women, while oropharyngeal neoplasms are the most prevalent among men [22,23].

### Brief History/Development of Urine HPV Testing

In conjunction with the extensive research which has been performed in the field of self-sampling of vaginal secretions for cervical screening, detection or HPV-related biomarkers in urine have also emerged as a reliable and less invasive approach than a traditional cervical screening [6,16]. As early as 2006, before the publication of the first meta-analysis on urine HPV assessment, Daponte et al. utilized polymerase chain reaction (PCR) methods and concluded that the detection of HPV in urine correlates well with the detection of concomitant cervical HPV even during pregnancy, while also stressing the importance of testing first-void urine (FVU) and quantifying viral load levels [12,13,14,19].

Later, in 2014, the first important meta-analyses had been published; Pathak et al. conclusively regarded urine HPV testing as a feasible and acceptable alternative in established cervical screening modalities suggesting it as a possible complementary organized screening method [24]. Sensitivity and specificity had to be compared to traditional methods, and Arbyn et al. conducted a further accurate meta-analysis on self-samples, which concluded that hrHPV testing on patient obtained material was less sensitive than on clinician taken samples, attributing the lower sensitivity to the use of assays based on signal amplification, which addressed the issue that not all HPV methods and publications had equal accuracy when used for self-sampling [25].

In a subsequent meta-analysis by the same author on self-samples that included 56 accuracy studies and 25 participation trials, Arbyn et al. focused primarily on vaginal self-sampling and illustrated that hrHPV assays based on polymerase chain reaction were as sensitive on self-samples as on clinician samples to detect CIN2+ or CIN3+, with hrHPV assays based on signal amplification showing less sensitivity on self-samples (Table 1) [18]. Taking this parameter into account, this meta-analysis documented an only marginally lower specificity in excluding CIN2+ lesions for self-samples than for clinician obtained samples. In terms of acceptance and feasibility, this later 2018 Arbyn meta-analysis illustrated that mailing self-sample kits to the women generated higher response rates than invitations or reminder letters. Apparently, in under-screened communities, direct offers of self-sampling devices to women also generated high participation rates. The authors considered that pilots should be set up before regional or national rollout of self-sampling strategies to compensate for the variable response rates among settings [18].

Aiming to address the poor reproducibility of HPV-related assays in self-samples, Arbyn launched the VALHUDES protocol and encouraged focusing on the development of tailored molecular tests that allow reflex testing of hrHPV in self-samples [26]. According to these authors, hyper-methylation of certain viral or human genes involved in carcinogenesis might also represent good candidates. Since most studies on urine HPV document only virological outcomes, the authors prioritize the deployment of studies assessing the clinical accuracy of hrHPV testing on urine. Finally, they consider the enforcement of the primary care provider’s role in the correct design of self-sampling programs paramount, as also illustrated by other authors [6,16].

## 2. Recent Developments of Urine HPV Testing

Several authors have recently published their contributions in urine self-sampling. In this perspective, their early innovation with the development of a special apparatus for the scope of urine collection represented a mainstay development [29]. The pioneering group of Vorsters has repeatedly illustrated that optimizing urinary HPV DNA detection should involve: (i) use of first-void urine (FVU); (ii) prevention of human/HPV DNA degradation during extraction and storage by adding a preservative; (iii) processing of a sufficient volume of whole urine; and the (iv) use of an analytically sensitive HPV test/recovery of cell-free HPV DNA in addition to cell-associated DNA [30,31].

Additionally, the importance of HPV genotyping in HPV lesions, genital cancer, and also cervical screening with self-samples was increasingly emphasized [12,13,14,16,19,32,33,34,35,36].

A recent study by Oliveira et. al. assessed HPV16/18-E6 expression prior to the treatment of cervical lesions and evaluated the concordance between urinary, vaginal, and cervical HPV16/18-E6 and hrHPV DNA testing [27]. Overall, HPV16/18-E6 oncoprotein was detected in 30.6% of cervical samples, 20.3% of self-collected vaginal samples, and 21% of urine samples. As of clinical sensitivity, the HPV16/18-E6 oncoprotein was undetected in CIN2 cases but was detected at low rates in CIN3 cases. The clinical sensitivity of the HPV16/18-E6 oncoprotein assessment in the detection of invasive cervical cancer was 70% for cervical scrapes, 55% for self-collected vaginal samples, and 52% for urine samples. Interpreting their results, the authors suggest that E6 oncoprotein detection, which can be tested in urine samples, serves for the clinically important detection of invasive or microinvasive lesions (Table 1).

Methylation biomarkers, as suggested by Arbyn et al., represent a promising and attractive approach (Table 2) [18]. Van den Helder et al. tested a panel of five methylation markers (ASCL1, GHSR, LHX8, SST, ZIC1) in three different urine fractions: full void urine, urine sediment, and urine supernatant. Strong correlations (r > 0.60) were found between urine fractions and methylation levels, which increased significantly with aggravated severity of underlying disease in all three fractions, another clinically important result. Comparison of cancer to controls was highly significant for all markers in all fractions; however, urine sediment performed best to detect CIN3. The authors consider their results to justify the potential of CIN3 detection by urinary methylation analysis (Table 2) [37].

Two further recent studies have also been published by a research group that developed an ultrasensitive nanowire assay [38,39]. Both studies used polyethylenimine-conjugated nanowires (PEI-NWs) and implemented HPV DNA isolation and detection strategy in urine samples. The authors consider that this assay demonstrated excellent ability to identify HPV DNA from urine specimens, as they observed an excellent agreement in the detection of high-risk HPV between paired urine and cervical samples, even with a small urine sample volume (Table 2).

### 2.1. mRNA E6/E7

Three recent studies have assessed the performance of the APTIMA hrHPV E6/E7mRNA assay in urine. In the study of Asciutto et al., the sensitivity of the APTIMA HPV assay in detecting HSIL/AIS/cancer for the clinician-taken cervical HPV samples was 100.0%; however, the corresponding value for urine self-sampling was a disappointing 44.8% (Table 2) [40]. In their study, Arias et al. illustrated that treatment of first void urine with Aptima Transfer Solution optimized the detection of high-risk HPV E6/E7 mRNA with the APTIMA assay (Table 2) [41]. Finally, the paper of Padhy et al. focused on the detection of hrHPV E6/E7 mRNA expression in urine samples, comparing their concordance with cervical samples including HPV 16 & 18/45 genotyping, and determining the utility in detecting ≥CIN 2 lesions. In this study, the sensitivity of urine hrHPV E6/E7 mRNA detection was 31.5%, while the specificity and PPV were above 95%. In their concluding remarks, the authors consider that using the APTIMA assay, urine hrHPV-mRNA detection is suboptimal for cervical cancer screening, but given the high specificity, it has the potential to identify high-grade lesions (≥CIN 2). The authors also point that urine hrHPV-mRNA genotyping via this modality is not beneficial in triage settings of normal or abnormal cytology to determine the need for colposcopy (Table 1) [1].

### 2.2. HPV DNA

A recent Danish study investigated the concordance of hrHPV positivity of home-based collected urine samples with vaginal self-samples as well as with cervical cytology samples collected by a general practitioner by using two different hrHPV DNA assays (COBAS^®^ 4800 & GENOMICA CLART^®^ HPV4S). Urine samples showed good concordance in hrHPV detection compared with vaginal and cervical samples when assessed with the COBAS assay and moderate with CLART. Additionally, urinary hrHPV detection by COBAS illustrated a sensitivity of 63.9% and a specificity of 96.5% compared with 51.6% and 92.4%, respectively, for CLART. While underscoring that participating individuals considered home-based urine collection as well-accepted and ranked it as the most preferred future screening procedure, the authors urge for improvement in accuracy rates of urinary hrHPV detection before urine can serve as an alternative screening option (Table 1) [28].

Another recent Korean study aimed to evaluate the diagnostic accuracy of two PCR-based hrHPV assays (Realtime HR-S and Anyplex II) on self-collected vaginal and urine samples for the detection of cervical precancer in a colposcopy population. In this study, Cho et al. conclude that despite the comparable detection performance for hrHPV and CIN2+ on self-collected vaginal samples and clinician-collected cervical samples, the performance of these two particular hrHPV tests using urine was inferior to those using clinician-collected cervical samples in terms of detecting hrHPV and CIN2+ (Table 2) [42]. In the most recent study, Cadman et al. compared the performance of four vaginal self-sampling devices, including wet and dry transport methods between themselves and an initial stream urine sample using the Becton Dickinson Onclarity assay in 600 women referred for colposcopy. Urine samples were collected with the Colli-Pee^®^ device, and two paired vaginal self-samples were obtained subsequently, using either simple devices (a dry flocked swab (DF) and a Dacron swab (WD)), or more complicated inventions (a HerSwab (HS) and Qvintip (QT) device). Women harboring irregular colposcopic findings underwent cervical biopsies; for these patients, histology results were also available. The authors assessed for HPV positivity and user preference and confidence in the collection; they also conducted additional analyses to examine the effect of adjusting for sample cellularity and different positivity thresholds [43]. Summarizing their results, the authors did not report any significant difference for the vaginal samples in terms of hrHPV positivity rates, which were closely distributed in a narrow range (from 65.1–71.7%), while the corresponding value for urine was a comparable 76%. Agreement on the same sample between the simple devices WD and DF were high (kappa = 0.801, 95% CI: 0.777, 0.826) but lower between the more complicated QT and HS (kappa = 0.753, 95% CI: 0.723, 0.779) and lower still between urine and the four vaginal samples (kappa = 0.568–0.646), largely due to the higher positivity rate with urine. There were no statistically significant or obvious differences in genotype-specific positivity by device. The highest sensitivity for CIN3+ was obtained with a simple and cheaper device (WD = 91.2%). Women found urine easiest to collect, and they were more confident that they had obtained the sample correctly. The authors conclude that both urine and vaginal self-samples obtained with simple devices performed well and were well-received by women (Table 2) [43].

The recent study by Östensson et al. that focused on a special population of patients who had previously undergone conization is of special interest. The authors studied 531 women and utilized the Abbott RealTime High-Risk HPV assay to examine how well the HPV findings from self-sampled vaginal (VSS) and urine specimens containing transport medium correctly identified women with recurrent CIN2+ compared to samples collected by clinicians. At the first follow-up approximately six months post-treatment, all patients with recurrent CIN3 had positive HPV results by all methods. At the second follow-up approximately one-year post-treatment, all seven newly-detected CIN2/3 recurrences were associated with HPV positivity on VSS and clinician-samples. Only clinician-collected samples detected HPV positivity for the two adenocarcinoma-in-situ recurrences, which were both HPV18 positive [44]. For overall HPV results, Cohen’s kappa revealed substantial agreement between VSS and clinician sampling and moderate agreement between urine and clinician sampling. Furthermore, clinician sampling and VSS were highly concordant for HPV16. For squamous pathology, VSS appeared as sensitive as clinician sampling for HPV in predicting outcome among the studied cohort (Table 2).

Finally, in a very recent Danish cross-sectional study in colposcopy referred individuals, Ørnskov et al. compared the absolute and relative sensitivity as well as the specificity of two self-collected specimens (Urine/Self-collected vaginal samples) and a clinician-taken Cervical Sample (CS) to detect high-grade cervical intraepithelial lesions (CIN2+/CIN3+), by using the Cobas HPV assay [Cobas 4800™ (Roche, Basel, Switzerland)]. They illustrated that both urine and vaginal self-collected sampling are non-inferior to the clinician-taken CS in identifying CIN2+/CIN3+ (Table 2). Furthermore, regarding the acceptability of the screening method used, they concluded that the majority of women identified urine as the sampling method of choice, especially when they were asked about the expected preference for women not attending the screening programme [17].

## 3. Urine HPV as A Proxy for HPV Vaccination Coverage

The assessment of HPV-related biomarkers detection in vaccinated populations is a continuing challenge [45,46]. In the emerging field of monitoring the results of HPV vaccination, detecting HPV antibodies in urine represents an alternative promising approach for noninvasive sampling to monitor HPV antibody status in women participating in large epidemiological studies and HPV vaccine trials [47]. Featuring early involvement in the topic, Vorsters’ group has updated their work. Summarizing the potential of these efforts, the authors conclude that the simultaneous assessment of both HPV infection and immunogenicity on a non-invasive, readily obtained sample indeed appears particularly attractive [47].

A recent study by Van Keer et al. investigated the properties of HPV-specific antibody transudates from systemic circulation in first-void urine of (un)vaccinated subjects and the agreement with paired sera [48]. Strong positive correlations were found in HPV-specific antibody levels between paired samples. In both first-void urine and serum, significantly higher HPV6/11/16/18 antibody levels were observed in vaccinated women compared with unvaccinated women (*p* ≤ 0.017). This study illustrated that vaccine-induced HPV antibodies are detectable in the first-void urine of young women, while significant positive correlations were documented between HPV6/11/16/18-antibodies in first-void urine and paired sera.

The second study compared the measurement of HPV antibodies in FV urine using a multiplex L1/L2 virus-like particles (VLP)-based ELISA (M4ELISA) with previously reported results using a glutathione S-transferase (GST)-L1- based immunoassay (GST-L1-MIA) [49]. As expected, lower HPV antibody concentrations were found in FV urine than in serum. Vaccinated women had significantly higher HPV6/11/16/18 antibody levels in both FV urine and serum compared with those unvaccinated; HPV antibody levels in FV urine and serum showed a significant positive correlation. Despite assay differences, there was a moderate to good correlation between M4ELISA and GST-L1-MIA. The authors consider that the comparable detection rates of FV urine HPV antibody with both assays further support this noninvasive sampling method as a possible candidate for HPV vaccine assessment.

## 4. Discussion

The non-invasive nature and favorable acceptability profile of urine HPV detection are most promising. However, currently, no HPV assay is specifically modified and marked for first-void urine, although devices to collect it have been developed and preservation mediums have been identified. Therefore, more investigation, focusing mainly on urine HPV DNA measurements with PCR-based methods and possible implementation in vaccination programs and HPV screening strategies, is required [50].

Several emerging possibilities for the implementation of urine HPV detection require further research. In the field of monitoring HPV vaccinations and primary genital and cervical cancer prevention, the assessment of uptake in HPV vaccination programs using proxy HPV antibody concentrations in FVU is an attractive approach and cost-effective policy as previously illustrated, especially in difficult to reach populations or special minorities [6,16].

In the quest for an accurate reflection of vaccination coverage in diverse populations, the choice of the particular assay to be used in HPV urine detection is important and deserves evaluation with carefully designed prospective trials. It must be emphasized that these candidate assays should be able to detect shifts in HPV genotype prevalence in the context of identifying possible type replacements in a cohort.

The assessment of urine HPV represents an invaluable tool that should not be confined to the primary vaccination cohorts of preadolescent girls, but it should be used for screening men and women of all ages.

Regarding urine self-sampling HPV tests, it is necessary for future research to describe the implementation status in the different parts of the world and the acceptance status of examinees in more detail, which will help design future screening strategies. In developing these screening strategies, it is desirable to consider the effect of using self-sampling HPV detection methods like urine sampling in the sufficiency of medical resources that can focus on the prevention of cervical cancer in countries around the world, including developing countries.

In the area of secondary cervical cancer prevention, the incorporation of urine HPV detection assays might vary depending on the settings. Studies have illustrated that primary cervical cancer prevention strategies can be individually deployed in various health systems and different target populations (a) maximal- and enhanced-resource settings, (b) limited-resource settings, and (c) basic-resource settings [2,9,17,51]. In a similar manner, secondary cervical cancer prevention strategies and triage using reliable biomarkers can be tailored and developed to meet the needs of the individual health system. In pragmatic terms, the deployment of screening strategies based on accurate urine HPV detection, which are capable of achieving high population coverage, while requiring limited costs and infrastructure, currently appears particularly attractive for limited and basic resource settings [17,32,34,46,52,53,54,55].

Patients attending colposcopy clinics who have been vaccinated following a local cervical treatment [52] or in the context of the follow-up of established lesions [45] represent additional potential candidates for the assessment of urine HPV with tailored tests. The utilization of sensitive urine HPV assays as a test of cure following local treatments in the cervix is also an attractive potential option [44,45].

## 5. Conclusions

Urine HPV detection due to the low costs inflicted by bypassing the doctors’ visit, the non-invasive nature, and the favorable acceptability profile has the potential to become the most promising tool that could expand the possibilities in a changing genital and cervical cancer prevention screening as well as in the surveillance and management of genital precancer and vaccination surveillance. The following points should be considered in clinical applications of HPV urine measurements or future trials; (i) Use of first-void urine (FVU) and purpose-designed collection devices; (ii) using a preservation medium to avoid human/HPV DNA degradation during the extraction and storage; (iii) using PCR-based assays, ideally with genotyping capabilities; (iv) processing of a sufficient volume of whole urine; and (v) the use of an analytically sensitive HPV test/recovery of cell-free HPV DNA in addition to cell-associated DNA.

## Figures and Tables

**Table 1 cancers-13-01640-t001:** Performance Characteristics of Urine HPV assessment in Routine Cervical Screening (Community). Studies focusing on General population patients.

Authors	Specimen	HPV Biomarker & Assay	Sensitivity	Cohen’s Kappa Coefficient (κ)
Oliveira et al. 2020 [27]	Paired urine s/s, vaginal s/s and cervical clinician-obtained	HPV16/18-E6 test vs. hrHPV DNA [OncoE6™ (Arbor Vita) vs. Cobas 4800™ (Roche)]	Cobas 4800™ Sensitivity for Cervical: 66.1% Cobas 4800™ Sensitivity for Vaginal: 65.3% (*p* = 1.00) Cobas 4800™ Sensitivity for Urine: 50.0% (*p* *<* 0.01) OncoE6™ Sensitivity for Cervical: 30.6% OncoE6™ Sensitivity for Vaginal: 20.2% (*p* *<* 0.01) OncoE6™ Sensitivity for Urine: 21.0% (*p* *<* 0.01)	Comparison between HPV-DNA and HPV16/18-E6 test results for types 16 and 18 according to sample origin Cervical: 0.76 Vaginal: 0.54 Urine: 0.55
Padhy et al. 2020 [1]	Paired urine s/s and cervical	Aptima mRNA™ (Hologic)	Urine hrHPV-mRNA detection: 31.5% Urine hrHPV-mRNA genotyping: 20.0%	Detection: 0.22 (*p* = 0.04) Genotyping: 0.25 (*p* = 0.16)
Tranberg et al. 2020 [28]	Paired urine s/s, vaginal s/s and cervical GP obtained	Pts with ASC-US CLART™ (Genomica) Cobas 4800™ (Roche)	Cobas 4800™: 63.9% urinary hrHPV detection compared to cervical sampling CLART™: 51.6% urinary hrHPV detection compared to cervical sampling	0.66 urine/vaginal concordance for hrHPV detection (Cobas 4800™) 0.55 urine/vaginal concordance for hrHPV detection (CLART™)
Pathak et al. 2014 [24]	Meta-analysis of 14 studies (1443 women) confined to urine s/s	Most used commercial PCR methods on FVU No independent analysis for PCR vs. signal amplification methods	Urine detection of any HPV: pooled sensitivity of 87% (95% C.I. 78% to 92%) Urine detection of hrHPV: pooled sensitivity of 77% (68% to 84%) Urine detection of HPV 16 & 18: pooled sensitivity of 73% (56% to 86%) Metaregression revealed an increase in sensitivity with FVU samples compared with random or midstream (*p* = 0.004)	

Abbreviations: self-sampled (s/s), polymerase chain reaction (PCR), first-void urine (FVU), and high-risk HPV (hrHPV).

**Table 2 cancers-13-01640-t002:** Performance Characteristics of Urine HPV assessment in Colposcopy Clinic patients. Studies focusing on Colposcopy patients.

Authors	Specimen	HPV Biomarker & Assay	Sensitivity (%)	Cohen’s Kappa Coefficient (κ)
Van den Helder et al. 2020 [37]	Full void urine, urine sediment & urine supernatant s/s	5 methylation markers Multiplex 1 (GHSR-SST-ZIC1)Multiplex 2 (ASCL1-LHX8)		
Lee et al. 2020 [38]	Unspecified populationCervical clinician-obtained vs. urine s/s	Cervical specimens: Cobas 4800™ (Roche) Urine: Ultrasensitive nanowire assay	HPV16: 81.3% (95% C.I. 54.4–96.0) HPV18: 100.0% (95% C.I. 29.2–100.0) Other hrHPV’s: 96.4% (95% C.I. 81.7–99.9)	HPV16: 0.83 (95% C.I. 0.67–0.99) Hpv18: 0.65 (95% C.I. 0.28–1.00) Other hrHPV’s: 0.97 (95% C.I. 0.91–100.0)
Asciutto et al. 2018 [40]	Clinician-obtained cervical cytology & mRNA HPV, Vaginal s/s, Urine s/s. vs. Cervical Histology (Biopsy or LEEP)	Aptima mRNA™ (Hologic)	APTIMA™ in vaginal self-samples 85.5% APTIMA™ in urinary samples 44.8% Cervical Clinician-obtained Cytology 81.7% Cervical Clinician-obtained APTIMA™ 100% for HG	
Arias et al. 2019 [41]	*Female arm:* Cervical cytology & Cervical histology, FVU s/s treated with proteinase K Untreated FVU s/s			*Arm A* Untreated FVU 0.29 (0.16–0.42) ATS-FVU 0.63 (0.52–0.73) *Arm B* Untreated FVU 0.27 (0.14–0.41) ATS-FVU 0.42 (0.30–0.54)
Cho et al. 2020 [42]	Matched urine s/s, vaginal s/s, and clinician-obtained cervical samples	RealTime HR-S HPV™ Anyplex II HPV 28™	For CIN2: Cervical samples, Realtime HR-S: 93.13% (95% CI, 87.36 to 96.81) Cervical samples, Anyplex II: 90.08% (95% CI, 83.63 to 94.61) Vaginal samples, Realtime HR-S: 84.73% (95% CI, 77.41 to 90.42) Vaginal samples, Anyplex II: 78.63% (95% CI, 70.61 to 85.30) Urine samples, Realtime HR-S: 73.28% (95% CI, 64.85 to 80.63) Urine samples, Anyplex II: 66.41% (95% CI, 57.61 to 74.42)	
Cadman et al. 2021 [43]	Urine & Vaginal s/s [*WD*=Digene^®^ Female Swab Specimen Collection Kit (Qiagen GmbH) placed in liquid specimen transport medium (STM) plus *DF* = Copan FLOQswab™ (Copan Diagnostics Inc) as a dry sample vs. the *QT* = Qvintip^®^ kit (Aprovix AB) & the *HS* = HerSwab (Eve Medical) both as dry samples]	hrHPV Onclarity™ (BD)	Sensitivity for CIN3+ with WD: 91.2% Sensitivity for CIN3+ with urine: 89.7% Sensitivity for CIN3+ with DF: 88.2% Sensitivity for CIN3+ with QT: 81.8% Sensitivity for CIN3+ with HS: 77.4%	Between WD & DF: kappa = 0.801, 95% CI: 0.777, 0.826 Between QT & HS: kappa = 0.753, 95% CI: 0.723, 0.779 Between urine and the 4 vaginal samples: kappa = 0.568–0.646
Pathak et al. 2014 [24]	Meta-analysis of 14 studies (1443 women) confined to urine s.s.	Most used commercial PCR methods on FVU No independent analysis for PCR vs. signal amplification methods	Urine detection of any HPV: pooled sensitivity of 87% (95% C.I. 78% to 92%) Urine detection of hrHPV: pooled sensitivity of 77% (68% to 84%) Urine detection of HPV 16 & 18: pooled sensitivity of 73% (56% to 86%) Metaregression revealed an increase in sensitivity with FVU samples compared with random or midstream (*p* = 0.004)	
Arbyn et al. 2018 [18]	Meta-analysis		Each hrHPV assay based on *PCR* was equally sensitive for CIN2+ and for CIN3+ on self-samples versus clinician samples. The pooled sensitivity of hrHPV assays based on *signal amplification* was 10–16% lower on a self-sample versus a clinician sample for all self-sampling device and storage medium categories	
Östensson et al. 2021 [44]	Pretreated patients with conizationPaired urine, vaginal s/s and cervical obtained by gynaecologist	Abbott RealTime High-Risk HPV PCR (Urine & Vaginal s.s, Cervical clinician-obtained)Cobas 4800™ (Roche) Cervical clinician-obtained	Clinician sampled: Any HR type:100% Clinician sampled: 16/18: 100% Vaginal self- sampled: Any HR type:100% Vaginal self- sampled: 16/18: 100% Urine self- sampled: Any HR type:100% Urine self- sampled: 16/18: 33%	Abbott clinician kappa = 0.83 Vaginal self-sample: kappa ranged from 0.63 to 0.89 Urine self-sample: kappa ranged from 0.36 to 0.85
Ørnskov et al. 2021 [17]	Paired urine, vaginal s/s and cervical obtained by gynaecologist	Cobas 4800™ (Roche)	Sensitivity for CIN2+ with Self-collected vaginal sample: 96% Sensitivity for CIN2+ with Urine sample: 93% Sensitivity for CIN2+ with Clinician-taken cervical sample: 96% Sensitivity for CIN3+ with Self-collected vaginal sample: 97% Sensitivity for CIN3+ with Urine sample: 95% Sensitivity for CIN3+ with Clinician-taken cervical sample: 97%	Between Urine and self-collected vaginal samples, kappa = 0.77, 95% CI: 0.70, 0.85 Between clinician-taken cervical samples and self-collected vaginal samples, kappa = 0.77, 95% CI: 0.70, 0.85 Between Urine and clinician-taken cervical samples, kappa = 0.66, 95% CI: 0.57, 0.74

Abbreviation: s/s = self-sampled.

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
