# Peer review of "Urine HPV in the Context of Genital and Cervical Cancer Screening—An Update of Current Literature"

_cancers, 2021, doi:10.3390/cancers13071640_

Round 1
Reviewer 1 Report
There is little hesitancy in recommending that this excellent review, entitled "Urine HPV in the context of genital and cervical cancer screening 2 – An update of current literature" be published. In so many respects, this review is remarkably up-to-date to include significant references to Y2020. Its discussion on the promise of non-invasive and acceptability of urine HPV detection is an important read. The presentation on recent urine HPV developments is very inclusive of current research on biomarkers to include mRNA E6/E7 and HPV DNA, which are highlighted.
Author Response
Authors’ actions: No action required, thank you. We have updated our references by including few most recent relevant and important studies.
Reviewer 2 Report
I read this topic with great interest. There are some comment to review.
Abstract>
Specific considerations on the five conditions presented at the end of abstract and conclusion should be presented in the text. There are duplicate keywords and too many.
Introduction>
The introduction is too long and there is a topic and related content, so overall deletion and correction are needed.
line 46: CxCa -> Unusual abbreviation and unnecessary abbreviation.
Line 53: It is understandable that screening is less likely due to the covid pandemic, but objective literature should be supported. If there is no literature, please delete it
line 62: suboptimal participation rates: it would be better to provide specific data or literature figures explaining how well optimal or suboptimal are.
Line 72~74: Adolescents and young women are not screening subjects for the convention cervical HPV test. Therefore, it cannot be screened for urine HPV test.
Line 79-82: There seems to be no reason to mention in this review the association between HPV and oropharyngeal cancer.
Line 95-116: Thought as a description of the history of the urine HPV test, it is too long. Separately, how about categorizing it under the subtitle History of Urine HPV test?
line 117: ~ goals In the context ~: In?
line 119-126: What is the meaning of the bold font?
2. recent development of urine HPV
line 127: how about add'test' at the end of title?
line 136-138: hardly understand this sentence.
I would like to suggest that summarize the results of the studies presented in topic 2. and present them in a single table.
line 208-230: It is a description of one study, but it is expressed in too many abbreviations, making it difficult to understand what the result of the study is about.
Discussion>
line 293: Colposcopy patients ~: what is the meaning?
Author Response
Comment 1: Abstract> Specific considerations on the five conditions presented at the end of abstract and conclusion should be presented in the text. There are duplicate keywords and too many.
Authors’ actions: Thank you for your remark. We have polished our language of our simple and main abstract of our manuscript. We have also deleted the duplicated keywords (Lines: 23-56).
Comment 2: Introduction> The introduction is too long and there is a topic and related content, so overall deletion and correction are needed.
Authors’ actions: Thank you for your remark. We have polished the language of the introduction (Lines: 57-105). Furthermore, we have added an extra section “Brief history/Development of HPV urine test” to demonstrate the evolution of the urine HPV testing approach (Lines: 106-148).
Comment 3: line 46: CxCa -> Unusual abbreviation and unnecessary abbreviation.
Authors’ actions: Thank you for your remark. We have deleted the above-mentioned abbreviation from lines 58, 347 where it has been used in the text. Additionally, all abbreviations have now been updated within the revised manuscript. Thank you (Lines 58 and 347).
Comment 4: Line 53: It is understandable that screening is less likely due to the covid pandemic, but objective literature should be supported. If there is no literature, please delete it.
Authors’ actions: Thank you for your comment. We have added 3 additional relevant references (Line 68 and 75).
Comment 5: line 62: suboptimal participation rates: it would be better to provide specific data or literature figures explaining how well optimal or suboptimal are.
Authors’ actions: Thank you for your comment. We have added a relevant reference (Line 77).
Comment 6: Line 72~74: Adolescents and young women are not screening subjects for the convention cervical HPV test. Therefore, it cannot be screened for urine HPV test.
Authors’ actions: Thank you for your comment. We have corrected this sentence (Lines 89-92).
Comment 7: Line 79-82: There seems to be no reason to mention in this review the association between HPV and oropharyngeal cancer.
Authors’ actions: Thank you for your comment. We have deleted the sentence according to your advice (Lines: 97-100).
Comment 8: Line 95-116: Thought as a description of the history of the urine HPV test, it is too long. Separately, how about categorizing it under the subtitle History of Urine HPV test?
Authors’ actions: Thank you for your remark. We have added an extra section “Brief history/Development of HPV urine test” to demonstrate the evolution of the particular HPV testing approach. See also answer to Comment 2. (Lines: 106-148)
Comment 9: line 117: ~ goals In the context ~: In?
Authors’ actions: Thank you for your remark. The sentence has been deleted and re-written (Lines: 138-143).
Comment 10: line 119-126: What is the meaning of the bold font?
Authors’ actions: Thank you for your remark. Bold fonds have been removed from the text.
Comment 11: 2. recent development of urine HPV. line 127: how about add' test' at the end of title?
Authors’ actions: Thank you for your point. The word “testing” has been added to the subtitle (Line 149).
Comment 12: line 136-138: hardly understand this sentence.
Authors’ actions: Thank you for your advice. Document numbered formats have been added to previous sentence (Lines: 155-158).
Comment 13: I would like to suggest that summarize the results of the studies presented in topic 2. and present them in a single table.
Authors’ actions: Thank you for your advice. We have now added 2 tables in our manuscript (Lines: 586-596).
Comment 14: line 208-230: It is a description of one study, but it is expressed in too many abbreviations, making it difficult to understand what the result of the study is about.
Authors’ actions: Thank you for your advice. Following your suggestion, the whole section has been reformatted. We hope that you find all amendments purposeful (Lines: 231-253).
Comment 15: Discussion> line 293: Colposcopy patients ~: what is the meaning?
Authors’ actions: Thank you for your point. The sentence has been re-written (Line 388).
Reviewer 3 Report
Reviewer Comments:
This review is about HPV test of urine samples related to cervical cancer prevention and is of scientific value, but it seems that the content of this review is not sufficient.
Specific critics are the following:
1) Regarding the sensitivity and specificity for detection of CIN2+/CIN3+, it is necessary to more comprehensively examine and clarify the three screening methods of clinician-taken samples, self-sampling, and urine samples.
2) Regarding self-sampling HPV test, it is necessary to describe the implementation status in the world and the acceptance status of examinees in more detail.
3) It is desirable to consider the degree of sufficiency of medical resources that can focus on the prevention of cervical cancer in countries around the world, including developing countries.
Author Response
Comment 0: This review is about HPV test of urine samples related to cervical cancer prevention and is of scientific value, but it seems that the content of this review is not sufficient.
Authors’ actions: Thank you for your comment. We have added the latest references.
Comment 1.1: Specific critics are the following: Regarding the sensitivity and specificity for detection of CIN2+/CIN3+, it is necessary to more comprehensively examine and clarify the three screening methods of clinician-taken samples, self-sampling, and urine samples.
Authors’ actions: Thank you for your comment. We added sensitivity and k values for clinician-taken samples, self-sampling, and urine samples in the text, where it was available and for “urine self-sampling” we added Tables 1 & 2.
- “Agreement on the same sample between the simple devices WD and DF was high (kappa = 0.801, 95% CI: 0.777, 0.826), but lower between the more complicated QT and HS (kappa = 0.753, 95% CI: 0.723, 0.779) and lower still between urine and the four vaginal samples (kappa = 0.568-0.646), largely due to the higher positivity rate with urine.”(Lines: 244-248).
- “For overall HPV results, Cohen's kappa revealed substantial agreement between VSS and clinician sampling, and moderate agreement between urine and clinician sampling.” (Lines: 293-295).
- Table 1 & Table 2. (Lines: 586-596).
Comment 1.2: Specific critics are the following: Regarding self-sampling HPV test, it is necessary to describe the implementation status in the world and the acceptance status of examinees in more detail.
Authors’ actions: Thank you for your remark. The information was not available in most of the papers reviewed as we stated in the revised discussion following your comment (Lines: 370-376).
Comment 1.3: Specific critics are the following: It is desirable to consider the degree of sufficiency of medical resources that can focus on the prevention of cervical cancer in countries around the world, including developing countries.
Authors’ actions: Thank you for your point. We have added this comment in the discussion (Lines: 377-387).
Round 2
Reviewer 3 Report
I express respect for the efforts of the authors who seriously responded to the strict question from the reviewer. I think this paper has been improved to increase its scientific value.